# Applications of Smart Textiles in Post-Stroke Rehabilitation

**DOI:** 10.3390/s20082370

**Published:** 2020-04-22

**Authors:** Ewa Korzeniewska, Andrzej Krawczyk, Józef Mróz, Elżbieta Wyszyńska, Rafał Zawiślak

**Affiliations:** 1Institute of Electrical Engineering Systems, Faculty of Electrical Engineering, Electronics, Computer and Control Engineering, Lodz University of Technology, 90-924 Lodz, Poland; 2Faculty of Transport and Computer Science, University of Economy and Innovations, 20-209 Lublin, Poland; ankra.new@gmail.com; 3Military Institute of Medicine, Clinic of Rehabilitation, 04-141 Warsaw, Poland; jmroz@wim.mil.pl (J.M.); ella.wysz@gmail.com (E.W.); 4Institute of Automatic Control, Lodz University of Technology, 90-924 Lodz, Poland; rafal.zawislak@p.lodz.pl

**Keywords:** electromagnetism in medicine, e-textile systems, smart glove, electrostimulation, stroke, hand, spasticity

## Abstract

Stroke is a disease affecting a large part of our society. According to WHO data, it is the second world’s biggest killer, accounting for near six million deaths in 2016 and it is about 30% of the total number of strokes per year. Other patients affected by such a disease should be rehabilitated as soon as possible. As a result of this phenomenon, paresis may occur. Among the devices available on the market there are many rehabilitation robots, but the method of electrostimulation can be used. The authors focused their attention on electrostimulation and commercially available therapies. Using this method, application to people with large hand muscle contracture is difficult. The authors of the work present a solution dedicated to exactly such people. A solution of textronic sensors manufactured on a textile substrate using the technology of physical vapor deposition is presented in the article. As a result of the conducted research, an electroconductive structure was obtained with a low surface resistance value of 1 Ω/□ and high flexibility. It can alternatively be used in hand rehabilitation for electrostimulation of fingertips. The solution is dedicated to people with high hands spasticity for whom it is impossible to put on a rehabilitation glove.

## 1. Introduction

Stroke is a neurological disease that affects mainly elderly people. Currently, the number of people suffering from the first strokes has increased approximately by 68% in Poland in the period time 1990–2016 and the number of disabled people is also increasing largely in the world.

The paresis of the upper limbs may be the result of ischemic stroke, which is the most common cause of permanent inability to work and perform basic life functions. Ischemic strokes account for over 80% of all strokes, while others are hemorrhagic strokes [1].

According to Saver [2], the cell destruction rate is 32,000 neurons per second, 230 million synapses per second and 200 m myelin fibers per second; the brain aging process is very high and equals 8.7 h/s. During a stroke in a minute, the human brain loses as many cells as in almost four years of normal life. For this reason, it is important to diagnose stroke early and, as a consequence, begin treatment immediately to reduce damage caused by tissue hypoxia (Figure 1). Ischemic stroke is caused by the stop or reduction of arterial perfusion. The applied treatment leads to its improvement in the ischemic area. Rehabilitation therapy that reduces the effects of stroke also has an impact on improving brain function [3,4].

Movement dysfunction is one of the key problems after stroke. Motor relearning during rehabilitation programs is based on repetition. Therefore, intensive and repetitive task-specific training, which involves the active practice of task-specific motor activities, is recommended after stroke. There are many different factors that have an impact on neuroplasticity after stroke. One of the most important factors on which the effectiveness of the therapy depends is the early implementation of treatment and rehabilitation. Individually-tailored rehabilitation to each patient with different abilities, functional needs and interests is the most important challenge in post-stroke care. The key time is limited to a critical period of six months. Due to the high plasticity of the brain, immediate and long-term rehabilitation allows reducing the neurological deficit. The restoration of brain functions is based on the distributed functional neural networks, which are located in the regions that are not touched by the brain infarction.

The next factors are regularity, repeatability, intensity of performed exercises. In the literature, we can find several methods that, when used together, bring the best results. Repetitive task training (RTT), Constraint-Induced Movement Therapy (CIMT), Strength Training, Aerobic Training, Non-Invasive Brain Stimulation (NIBS), Computer-Based Virtual Reality Training or Neuromuscular Electrical Stimulation.

Currently, we can also notice some major developments and innovations of Robotic-Assisted Rehabilitation (RR) and Robotic-Assisted Therapy (RT). From year to year, we observe an increase in the number of next-generation rehabilitation robots that can lead to significant benefits in neurorehabilitation. 

Using RT, we can measure several parameters that are usually not considered in conventional therapy. They include inter alia, spasticity, reflexes, level of voluntary control and functional movements. In addition, the therapy guarantees high intensity of repetitive movement training, task-specific and interactive treatment and monitoring a patient’s progress. Therefore, it is a perfect tool for doctors and therapists for facing neurorehabilitation challenges. Furthermore, robot-assisted therapy can also significantly decrease the labor intensity and costs associated with recovery training [5].

The rehabilitation assisted by robots and mechanical devices, because of their intended use, can be divided into two groups supporting the lower- and upper-limb rehabilitation. 

Some of them like the Robot RRH1, created in the Institute of Automatic Control Technical University of Lodz, may be used even for people lying in beds [6]. Research from recent years proves that the recovery process after stroke, modulated and strengthened by properly carried out rehabilitation, in the case of lower limb dysfunction may lead to partial or even complete recovery of the ability to move independently.

The second group is the therapy of the paresis of the upper limb. That one is considered the most difficult part of the process of rehabilitating patients after a stroke. We can divide the problems into two groups. One is a decrease in muscle tone (plegic) usually appearing immediately after a stroke, the other is a permanent limb cramp (spasticity) which is a consequence of giving up hand exercises when there is a decrease in muscle tension and it prevents any precise movement of the hand. [7].

Although many robotic rehabilitation devices have been developed and introduced at the market, e.g., [8], there is still a lot to be achieved. Rehabilitation equipment should fulfill core requirements such as: low cost, portability, extent of operation and tele-operation [9].

Another common problem is hand rehabilitation. High finger mobility allows us to grab, pinch, manipulate and interact with objects. Good, efficient hands without paresis are the basis for normal functioning in the environment in daily life. Additionally, in that case, to prevent permanent spasticity, post-stroke therapy should be carried out immediately after stroke. In the literature, we can find descriptions of many devices created for hand therapy of people with paresis [10,11,12]. Overall, we can divide them by construction into stationary and mobile devices [13]. Stationary devices available on the market, such as “Hand of Hope” by Reha-Robotics or AMADEO [14] are complex and can perform exercises based on the anti-spastic model. Unfortunately, these types of devices must be under the control of qualified personnel. 

The second group is based on a small size and portability solution of the system, which could allow the users to undergo rehabilitation on their own in their homes. Designed small robotic manipulators, so-called intelligent glows, mechanically supported tools to straighten and bend the hand, can decrease physiotherapist’s involvement during patients’ demanding treatment. Using tools like that can result in increasing the treatment quality and the number of patients rehabilitated at the same time. These popular devices are intended for people with neurological and orthopedic injuries resulting from stroke, Parkinson’s disease, musculoskeletal disorders and upper limb injuries. The system relies on the neuroplastic abilities of nerve tissue to create new connections and self-repair by stimulating the proprioceptive system. We can find many examples of mechatronic systems like the “Wearable hand rehabilitation system with force feedback” project being developed at the Institute of Automation and Robotics, Poznan University of Technology (Figure 2), based on two arms equipped with the latest generation force sensors and driven by a digital servo controlled by embedded ARM controllers or solutions based on actuators using McKibben’s artificial muscles construction [15]. In all cases, their operating principle is similar, bending or straightening, setting the fingers in motion. They are small, lightweight, easy to be used and give the ability to do most rehabilitation exercises. 

It is very important for stroke patients to improve the quality of hand function. In many cases, patients after stroke have post-stroke spasticity. Upper-limb rehabilitation is not sufficient to restore hand function. There is a solution on the market in the form of a textronic glove developed by scientists from the Institute of Neuroinformatics (Laboratory of Neural Plasticity) of the University of Bochum, Germany. According to their solution, a patient with hand paresis can be rehabilitated using a TipStim© glove. However, this solution is not satisfactory for people with high spasticity, for whom it is impossible to put on a glove (Figure 3).

The authors of the work propose an innovative solution consisting in the production of electric sensors on a composite elastic textile substrate, which can be used in an existing system of textronic gloves.

Different types of diagnostic sensors can be integrated into textiles with such methods such as: embroidery, physical vapor deposition, sputtering, chemical vacuum deposition, sol–gel, inkjet printing or polymer-assisted physical vapor deposition [16,17,18]. It should be noticed that all flat textiles are characterized by anisotropy of the electrical properties [19,20] and modification of electrical parameters of the sensors is possible with laser ablation of electroconductive thin layer [21,22,23]. The homogeneity of the created structure can be checked with a non-invasive method like optical coherence tomography [24,25] or thermography [26]. 

The percentage of e-textile players that use various technologies in the production of textronic solutions is shown in Figure 4 [27].

Wearable electronics solutions include sensors using changes in electrical parameters, such as impedance [28], inductance [29], capacitance [30], resistance [31], triboelectric phenomena [32,33], as well as thermoelectric [34], piezoresistive [28] piezoelectric [35], photoelectric [36] or optical [28]. They support the solving of society’s problems in the field of health care, work safety or comfort. Among many solutions of connecting electronic systems to conventional textiles, the electroconductive threads are used to produce woven, knitted or embroidered elements. The structure of threads intertwined has better flexibility and extensibility than the individual fibers. Connections of wearable electronics with textile substrates can also be made by laminating printed circuit assemblies on fabrics [37]. Embroidery using electroconductive yarns to produce electrical circuits and electrodes on textiles is described in [33]. Post et al. described the patterning of conductive textiles by numerically controlled sewing or weaving processes. They discussed different types of conductive yarns in which linear resistance was about 10 Ω/m. Silk fibers with a polymer coating were presented by Irwin et al. [38]. The threads were integrated into the fabric and the functional timer circuits were built. The tested fibers were characterized by much greater elasticity in comparison with rigid metal yarns constructed on the basis of steel. The conductivity of 8.5 S/cm was achieved. The embroidery technique using electroconductive threads has also been used by Roh [39] to develop a connection between different textile layers using a commercial computer numeric control embroidery machine. 

Many of the wearable sensors are in direct contact with human skin. For this reason, they should provide wearing comfort as for example the ACC/PAA/alginate hydrogel-based ionic skin sensor developed by Lei et al. which can be applied to detect gentle finger touch, complicated muscle movements during speaking, detect human motion or blood pressure [30]. The pressure sensor for health and tactile touch monitoring developed by Gao et al. [31] has also direct contact with the skin. Such a sensor, combined with a polydimethylsiloxane (PDMS) wristband, provides real-time information about the user’s pulse and, when placed in a PDMS glove, provides comprehensive tactile feedback of human hand touching. Some sensors used to detect human vital factors are created on electrospun nanofibers mats or on a non-textile flexible substrate such as Kapton [32]. Wearable sensors need also to be self-powered devices. It is possible to find the results of research on various energy harvesters or the triboelectric nanogenerators (TENGs). Parida et al. [35] developed a 3D-printable, highly conductive, extremely stretchable and healable TENG. The developed nanogenerator shows good conductivity (6250 S/cm) and record-high initial stretchability of 2500% and recovered 96.0% of its conductivity after healing. Some other examples of wearable energy harvesting devices constructed based on the thermoelectric phenomenon with desired dimensions are discussed in [34]. Another group worked on the sensors which monitored the perspiration lactate and illuminance and at the same time the energy harvester that extracted the electronic energy from the wearer’s metabolic production and photoenergy from ambient illumination [36].

Such systems can be used in various medical tests, such as electrocardiography (ECG), electromyography (EMG) and electroencephalography (EEG), pH level, temperature, blood pressure, breathing and many others. The multi-perspective view on patient data is necessary to improve medical decision making [40]. All these diagnostic measurements improve health care as they can be done online and the results can be wirelessly transmitted to the health center where the diagnosis on the patient’s state is undertaken. The e-mail sensors can be also used in the system of alarming if the state of the patient is somehow non-typical or even dangerous. It concerns infants and aged people mainly but also handicapped people. Thus, the area of applications is wide and it is probable that the society requirements will be bigger and bigger [41,42,43,44,45,46,47].

## 2. Materials and Methods

### 2.1. The Modified Solution

The new approach in electric stimulation using a wearable system has been mastered very recently. It is the electric stimulation of tips. The treatment is realized using two-directional pulse currents with a long interval time between pulses. It is assumed that the stimulation of tips evokes some change in neuronal cells in the patient brain. Technically, the stimulation is realized by a five-finger glove with electrodes installed on the tips (Figure 5). 

The glove is equipped with two independent electric circuits. One of them is used to stimulate the median nerve. In this case, the electrodes are placed on the distal and middle phalanges of fingers 1–3. The second electrical circuit applies to 4–5 fingers and is used to stimulate the ulnar nerve (Figure 6). The values of the current used for stimulation are in accordance with the recommendations of the manufacturer and the rehabilitation doctor.

### 2.2. Substrate

After long-term research and searching for a material that could act as a base for the stimulation electrode, an Optisana© bandage was chosen. It is produced on the basis of transparent nonwoven fabric which is made of loosely braided polyurethane threads. A thin electroconductive layer was produced on that substrate in the process of physical vapor deposition. The parameters of the metallic layer allow for implementing it into the TipStim glove system

### 2.3. Microscopic Research

All microscopic images were collected with the optical KEYENCE VK-X1000 microscope. The magnification of the tested structures is marked in the pictures.

### 2.4. Physical Vapor Deposition

A thin electroconductive layer was created in the process of physical vapor deposition using the Classic 250 vacuum chamber of the Pfeiffer Vacuum system. The process started after reaching 5 × 10^−5^ mbar initial vacuum and lasted 5 min. Silver with a high conductivity value (γ = 62.5 m/(Ω·mm^2^)) and high purity provided by Polish Mint Ltd. was used as a deposited material. The vacuum deposition process was carried out using a resistive resistance source made of tungsten, characterized by a much higher melting temperature than silver. The distance between the evaporation source and the surface on which the metallic conductive layer was deposited was 6 cm. Increasing this distance resulted in the inability to obtain a continuous layer with a low resistance value. Reducing the distance resulted in the patch overheating. Prior to the technological process, fibrous substrates were pre-conditioned for several hours at room temperature and 55% humidity. The connecting wires were attached to the sensor structure using ELPOX AX 15S electrically conductive glue (manufactured by AMEPOX, Łódź, Poland) with low resistivity (1.6 × 10^−6^ Ωm). Contact modification can be performed using laser ablation of its surface [22,23,24].

## 3. Results and Discussion

### 3.1. The New Solution

In the first stage of research, the surface structure of the electrodes used in the existing solution for rehabilitation was analyzed, as well as the structure of fibrous material present in the bandage, and used as a substrate to produce the median and ulnar nerves’ electrostimulation electrode. An image of the surface of the material used in the rehabilitation glove as an electrode is shown in Figure 7a (magnification 50×). The thread weave is visible; it is characteristic of a one-sided braided weave. Densely woven threads with metallic features are visible. A three-dimensional structure is visible in the photo. In Figure 7b (magnification 300×), loosely braided nylon threads are visible without significant ordering, but also with a three-dimensional arrangement. The presented geometry of the medical patch is not propitious to the production of structures with continuous electroconductivity.

Despite the lack of an organized weave between the threads in the chosen medical patch, a continuous metallic layer with a surface resistance of 1 Ω/□ was created as a result of physical vapor deposition. This resistance was measured in a typical four-electrode system [17].

The structure and surface profile of the embedded surface are shown in Figure 8.

The analysis of the microscopic image shows that, despite significant unevenness on the surface of the fibrous substrate, a thin-film structure with electrocontinuity was created. Atoms of stochastically deposited metal penetrated deep into the material creating a thin metallic layer. The roughness profile shown in Figure 8 indicates a very large surface unevenness. The plotted curve shows the surface profile marked in the figure as the blue line between the two red markers. The distance between the highest and lowest point of the structure (referred to as *R*_max_) is 332.74 µm [48]. Using physical vapor deposition, the production of an electrically conductive layer was successfully completed while maintaining the flexibility of the electrode. This is the advantage of the technology used to manufacture thin-film structures compared to, for example, inkjet printing, where structures using the electroconductive inks are produced by soaking the substrate. With too much metal deposited in the dispersion liquid, the flexibility of the resulting structure is limited. A photo of the patient’s hand undergoing electrostimulation using the developed electrodes is presented in Figure 9.

### 3.2. Case Study

A 36-year-old patient was admitted to rehabilitation and was admitted to the Rheumatology Clinic due to persistent paresis of the left upper limb, slight paresis of the lower limb. The patient was confirmed ischemic stroke of the right hemisphere during MRI magnetic resonance imaging. After the implementation of treatment and stabilization of the patient at the Rheumatology Clinic of the Military Medical Institute, she was redirected to rehabilitation treatment at the Rehabilitation Clinic. Then, after diagnosing deep sensation disorders while having preserved superficial sensation, she was qualified for electrostimulation using a TipStim glove.

Rehabilitation was carried out using the PNF (Proprioceptive Neuromuscular Facilitation) method for 120 min six times a week and, additionally, the fingertips were electrostimulated using a TipStim glove for 60 min with a frequency of once a day excluding Sundays for four weeks. After this period, the patient was again subjected to dexterity tests. The applied functional hand tests were aimed at an objective assessment of small and large motor functions of the hands using simulated everyday activities as well as muscle strength and the return of superficial and deep sensation. In addition, a comparison was made of the outcome of the patient’s therapy after a stroke resulting from systemic vasculitis with a healthy person of a similar age and gender. 

After four weeks of rehabilitation of a stroke patient the use of fingertip electrostimulation, significant motor limb improvement with paresis was obtained (Table 1). It confirms the correctness of the using electrostimulation even of small muscle groups to improve the mobility of patients after stroke.

## 4. Conclusions

Among the many technical solutions supporting the rehabilitation of people after stroke who have spasticity of the hands, solutions dedicated to supporting the restoration of the motor functions of the hands and palms, should be noted. In some patients, the level of paralysis of the nerves of the hand is so great that it is impossible to take advantage of the solutions offered by robotics. In this case, the solution adequate to such applications is rehabilitation using the TipStim glove with newly developed electrodes for electrostimulation. Thanks to the flexible substrate used as well as the process of physical vapor deposition, the produced electrodes are flexible and allow sensors to be worn by a person with any degree of dexterity of the hand.

In order to use electroconductive threads available on the market to create electrodes for finger electrostimulation, such a solution should be designed at the initial stage of glove design. Creating electrodes for electrostimulation of fingers based on dressing patches will allow personalizing the electrodes and adapt them to the needs of the patient. The electrodes produced in the PVD process will be irreplaceable in the situation of adjustment to the individual needs of the patient taking into account the size of the fingers. Using the proposed technology for producing a thin electroconductive layer on the proposed substrate, it is also possible to produce electrode matrices of the desired dimensions in one technological process by masking or laser ablation of the metal from the resulting structure.

In addition, it should be emphasized that no continuous electrical conductivity of the produced structures on textile substrates characterized by spatial structure is observed. In the described case, in the technological process of physical vapor deposition, obtaining thin electroconductive layers on such substrates was successful.

The authors of this study suggest using the method of electrostimulation of fingertips using the TipStim glove with the proposed new electrodes in the rehabilitation of patients with paresis of the upper limb after a stroke in order to improve superficial and deep sensation and improve the efficiency of the hand. The legitimacy of its use was confirmed by the therapies carried out for people with paresis. 

## Figures and Tables

**Figure 1 sensors-20-02370-f001:**
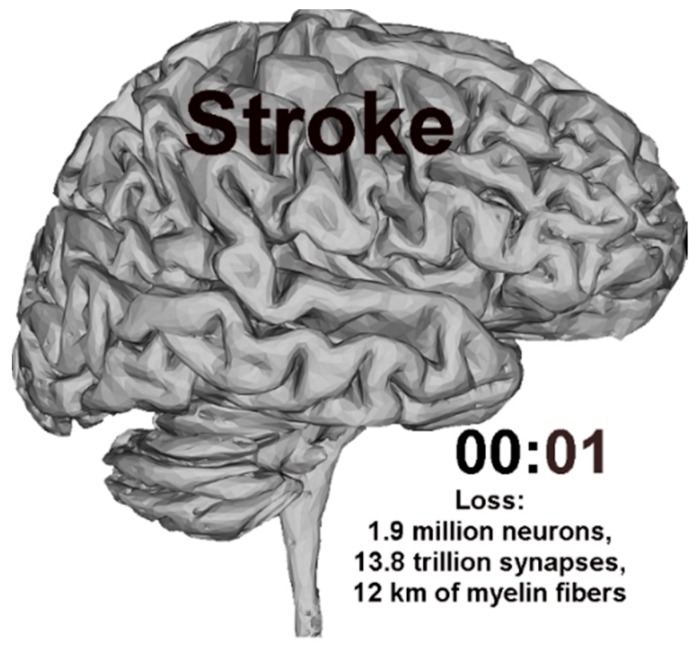
Average number of lost cells during one minute of stroke.

**Figure 2 sensors-20-02370-f002:**
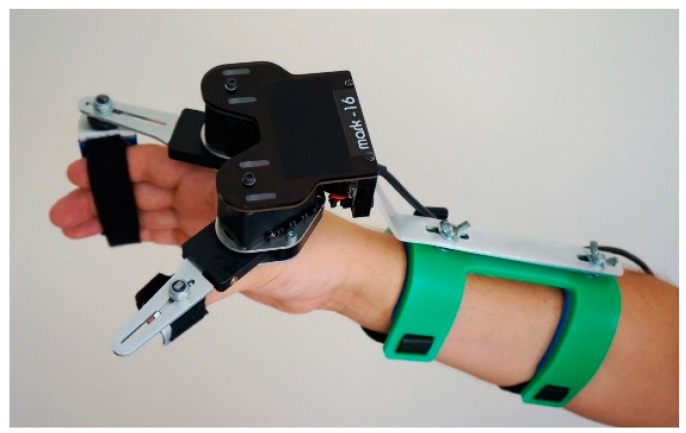
Example of a rehabilitation hand robot constructed in Poznan University of Technology.

**Figure 3 sensors-20-02370-f003:**
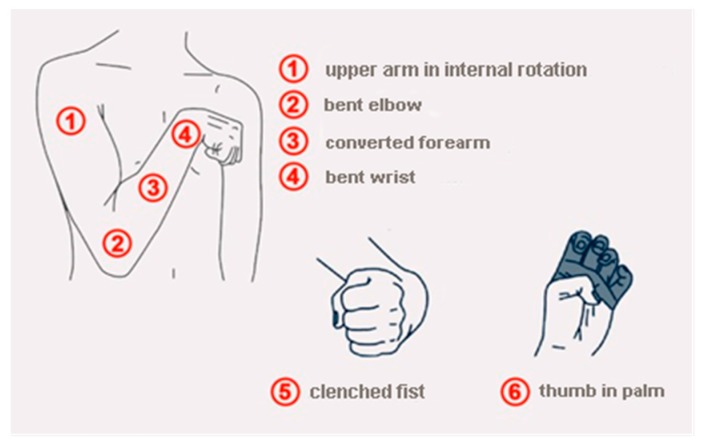
Characteristic features for post-impact spasticity.

**Figure 4 sensors-20-02370-f004:**
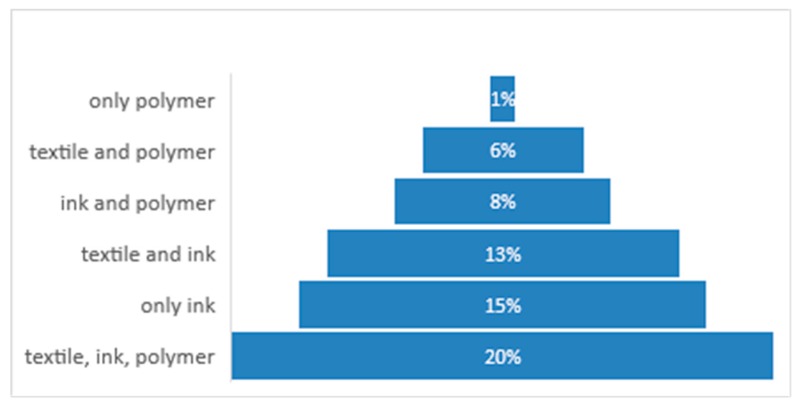
The percentage of e-textile players that use various technologies.

**Figure 5 sensors-20-02370-f005:**
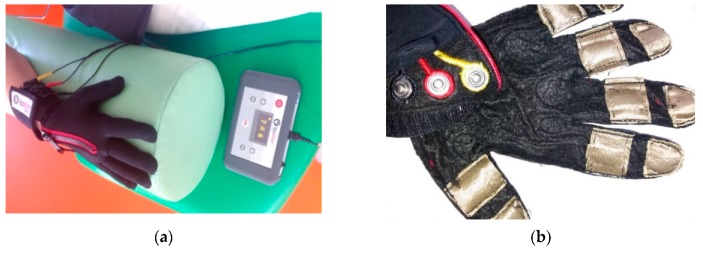
Electrostimulation therapy kit using the TipStim glove: (**a**) medical stand; (**b**) the internal view of stimulation glove.

**Figure 6 sensors-20-02370-f006:**
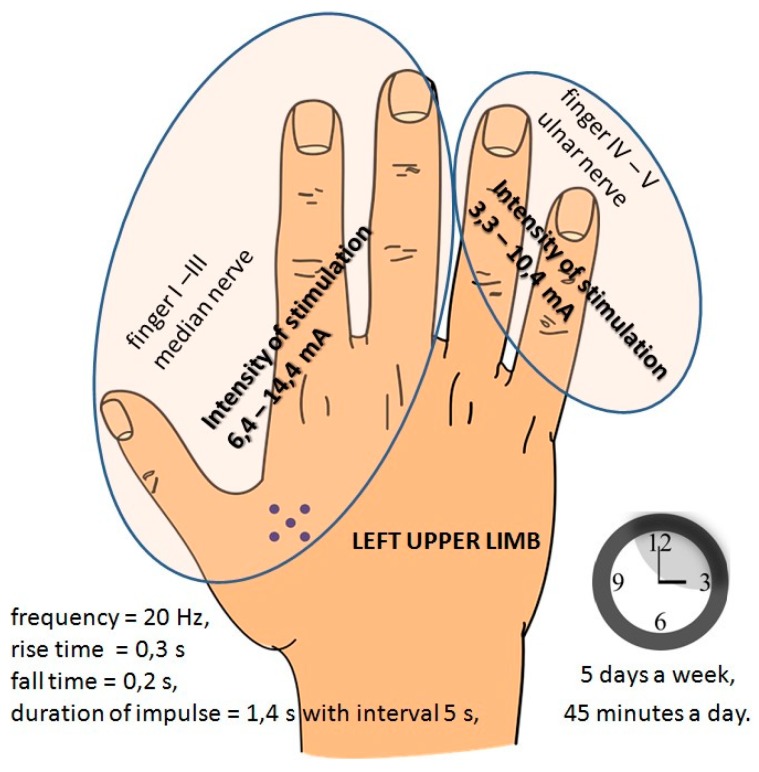
Neural connections between particular fingers and human organs and parameters of the stimulation.

**Figure 7 sensors-20-02370-f007:**
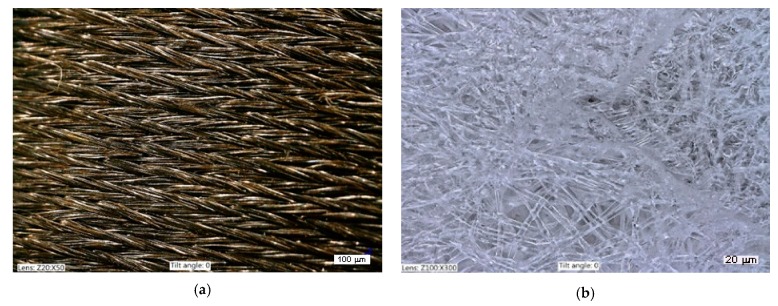
Microscopic images of textile electrodes (**a**) used in the modified TipStim solution. (**b**) Fibrous substrate of the Optisana patch.

**Figure 8 sensors-20-02370-f008:**
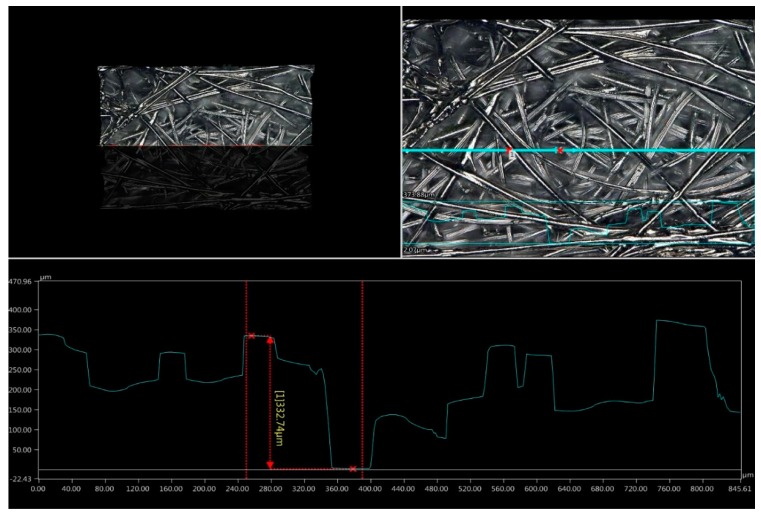
The structure and surface profile of the textile electrode produced in the PVD process on a fibrous substrate.

**Figure 9 sensors-20-02370-f009:**
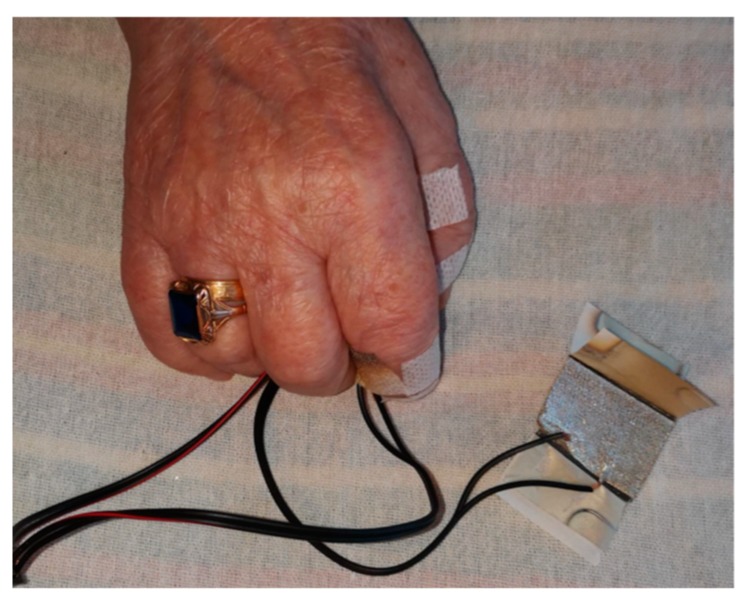
The alternative electrodes placed on the patient’s hand.

**Table 1 sensors-20-02370-t001:** Results of patient tests.

Test	Items Used for the Test	Right Hand			Left Hand		
		0 Week	2 Weeks	4 Weeks	0 Week	2 Weeks	4 Weeks
Writing	Pen and piece of paper A4	8.16 s	7.46 s	5.71 s	-	42.5 s	17.58 s
Flipping cards	Playing cards	5.11 s	4.1 s	3.5 s	-	25.5 s	8.08 s
Collecting coins into containers	Coins of different sizes	5.2 s	5.2 s	4.94 s	-	31.3 s	17.2 s
Collecting beads with a spoon into a container	Yogurt cup and 5 pieces of beads/chopping board	7.56 s	6.92 s	7.22 s	-	24.1 s	12.02 s
Stacking coins in a pile	5 pcs of 2 PLN coins	4.99 s	4.49 s	5.68 s	-	-	16.1 s
Moving objects to the designated field	Yogurt jars	3.15 s	3.22 s	3.28 s	14.5 s	9.37 s	6.5 s
Muscular strength handshake	Dynamometer	26 kg	29.3 kg	32 kg	2.3 kg	6 kg	12.7 kg

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
