# Peer review of "Applications of Smart Textiles in Post-Stroke Rehabilitation"

_sensors, 2020, doi:10.3390/s20082370_

Round 1

Reviewer 1 Report

The content of the paper is significant and highlights a vital need to develop potential home based rehabilitation tools to elevate the burden on the public health systems. The results were well presented and I am sure this subject will be of good interest to readers. The current virus situation had highlighted the fragility of the health systems and it is imperative to develop such potential products so patients and carers can take more ownership of their rehabilitation progress if they cannot access medical help for various reasons.

The material used for the gloves are of particular interest. There are conductive material that are commercially available, what are the advantages of creating the conductive material? 

What are the advantages of coating as compared to weaving or knitting yarns into the textiles? Production time and cost implications?

Author Response

Dear Reviewer,

We would like to thank you very much for your time you spent for correcting our paper and suggestions. They have helped us to improve the paper.

Below we have written answers for your comments. All of them are placed directly under the comments and they are highlighted in the text:

The content of the paper is significant and highlights a vital need to develop potential home based rehabilitation tools to elevate the burden on the public health systems. The results were well presented and I am sure this subject will be of good interest to readers. The current virus situation had highlighted the fragility of the health systems and it is imperative to develop such potential products so patients and carers can take more ownership of their rehabilitation progress if they cannot access medical help for various reasons.

The material used for the gloves are of particular interest. There are conductive material that are commercially available, what are the advantages of creating the conductive material?

The electrodes produced with the described method were constructed based on a commonly available dressing patch, on which a thin electroconductive silver layer was applied with the usage of physical vapor deposition technology. The use of this method allowed to maintain the elasticity of the substrate and the functional features of patches which are available on the market. To use conductive material that are commercially available, another method should be developed to make it possible to attach these materials to the fingers. Dressing patches that have been previously manufactured have such functionality. In addition, it should be emphasized that no continuous electrical conductivity of the produced structures on textile substrates characterized by spatial structure is observed. In the described case, in the technological process of physical vapor deposition, obtaining thin electroconductive layers on such substrates was successful.

The information presented above was placed in the manuscript in the Conclusions section.

What are the advantages of coating as compared to weaving or knitting yarns into the textiles? Production time and cost implications?

On the market the conductive materials are available. The conductive structures can be created also in PVD process. These are two completely different methods of producing electroconductive structures. In order to use electroconductive threads available on the market to create electrodes for finger electrostimulation, such a solution should be designed at the initial stage of glove design. Creating electrodes for electrostimulation of fingers based on dressing patches will allow to personalize the electrodes and adapt them to the needs of the patient.

At the stage of development of the technology for producing electrodes based on patches, the economic justification for their manufacture was not analyzed. The authors' goal was to fill the gap in access to rehabilitation for people with high spasticity of the hand. No calculations have been made so far, however, it should be expected that the braided thread method can work in the conditions of unified mass production, while the electrodes produced in the PVD process will be irreplaceable in the situation of adjustment to the individual needs of the patient taking into account the size of the fingers. Using the proposed technology for producing a thin electro conductive layer on the proposed substrate, it is also possible to produce electrode matrices of the desired dimensions in one technological process by masking or laser ablation of the metal from the resulting structure.

The information presented above was placed in the manuscript in the Conclusions section.

Attached please find the improved and corrected text of our article.

Reviewer 2 Report

The paper presents a conductive textile made by Ag evaporation and further used for electrostimulation of fingers for post-stroke rehabilitation in heavy cases when gloves cannot be used. One rehabilitation case is described.

An extensive context scope is given for many rehabilitation techniques mostly unrelated to the described work. Rehabilitation techniques are interesting but far from the review scope. This part should be shortened.

The electrostimulation itself is briefly described and does not make enough references to similar work (samples given below, directly related to the topic but lots of refs are available for conductive textiles). This part should be extended and amended.

---

Some detail comments:

The 1st sentence of the abstract does not seem correct or at least ambiguous: does stroke really affects 70% of people ?

line 119: grammar "actuators used the silicone"

line 129: GUI

Fig. 7b has no scale

Ref 16 is not available: the issue 16 is missing on the editor's website

Ref 16 is written in polish so the background of the team's work (basic ex-situ test procedure) is not available to english or non-polish readers.

---

The proposed work is in between engineering and clinical trial.

It's suitable for the review but provided a major rewriting of the text to focus more on the sensing and use of the sensor aspects: ex-situ characterization could be described ...

Sample refs to related work:

D. C. Irimia, M. S. Poboroniuc, S. Hartopanu, D. Sticea, G. Paicu and B. E. Ignat, "Post-stroke hand rehabilitation using a hybrid FES-robotic glove," 2016 International Conference and Exposition on Electrical and Power Engineering (EPE), Iasi, 2016, pp. 356-359.

Kattenstroth, J.C., Kalisch, T., Sczesny-Kaiser, M. et al. Daily repetitive sensory stimulation of the paretic hand for the treatment of sensorimotor deficits in patients with subacute stroke: RESET, a randomized, sham-controlled trial. BMC Neurol 18, 2 (2018). https://doi.org/10.1186/s12883-017-1006-z

Author Response

Dear Reviewers,

We would like to thank you very much for the time you spent on correcting our paper and suggestions. They have helped us to improve the paper.

Below we have written answers for your comments. All of them are placed directly under the comments and they are highlighted in the text:

The paper presents a conductive textile made by Ag evaporation and further used for electrostimulation of fingers for post-stroke rehabilitation in heavy cases when gloves cannot be used. One rehabilitation case is described.

An extensive context scope is given for many rehabilitation techniques mostly unrelated to the described work. Rehabilitation techniques are interesting but far from the review scope. This part should be shortened.

The mentioned part of the Introduction section has been shortened.

The electrostimulation itself is briefly described and does not make enough references to similar work (samples given below, directly related to the topic but lots of refs are available for conductive textiles). This part should be extended and amended.

The suggested references have been added to the text.

---

Some detail comments:

The 1st sentence of the abstract does not seem correct or at least ambiguous: does stroke really affects 70% of people ?

The mentioned value was taken from the literature, but we agree with the reviewer that this value is very huge, so we decide to change the sentence to: „Stroke is a disease affecting large part of our society

line 119: grammar "actuators used the silicone"

 this part of the manuscript was deleted

line 129: GUI

this part of the manuscript was deleted

Fig. 7b has no scale - 

It is corrected

Ref 16 is not available: the issue 16 is missing on the editor's website –

there was the mistake in the paper description, we have corrected it and to make it easier to find the article we have added the doi numer doi: 10.1515/phys-2018-0007

Ref 16 is written in polish so the background of the team's work (basic ex-situ test procedure) is not available to english or non-polish readers.

We think that the reviewer meant reference number 17. We remove the paper from the reference list.

---

The proposed work is in between engineering and clinical trial.

It's suitable for the review but provided a major rewriting of the text to focus more on the sensing and use of the sensor aspects: ex-situ characterization could be described ...

According to the reviewer’s suggestion Introduction of the manuscript was rewritten. The text about the rehabilitation techniques was shortened and the additional information focused on sensors was added to the manuscript:

Wearable electronics solutions include sensors using changes in electrical parameters, such as impedance [28], inductance [29], capacitance [30], resistance [31], triboelectric phenomena [32,33] as well as thermoelectric [34], piezoresistive [28] piezoelectric [35], photoelectric [36] or optical [28]. They support the solving of society's problems in the field of health care, work safety or comfort. Among many solutions of connecting electronic systems to conventional textile, the electroconductive threads are used to produce a woven, knitted or embroidered elements. The structure of threads intertwined has better flexibility and extensibility than the individual fibers. Connections of wearable electronics with textile substrates can also be made by laminating printed circuit assemblies on fabrics [37 Shi 2019]. Embroidery using electroconductive yarns to produce electrical circuits and electrodes on textiles is described in [33]. Post et al. described the patterning of conductive textiles by numerically controlled sewing or weaving processes. They discussed different types of conductive yarns which linear resistance was about 10 Ω/m. Silk fibers with a polymer coating were presented by Irwin et al [38]. The threads were integrated into the fabric and the functional timer circuits were built. The tested fibers were characterized by much greater elasticity in comparison with rigid metal yarns constructed on the basis of steel. The conductivity of 8.5 S/cm was achieved. The embroidery technique using electroconductive threads has also been used by Roh [39] to develop a connection between different textile layers using a commercial computer numeric control embroidery machine.

Many of the wearable sensors are in direct contact with human skin. For this reason, they should provide wearing comfort as for example the ACC/PAA/alginate hydrogel‐based ionic skin sensor developed by Lei et al which can be applied to detect gentle finger touch, complicated muscle movements during speaking, detect human motion or blood pressure [30]. Pressure sensor for health and tactile touch monitoring developed by Gao et al [31] has also the direct contact with the skin. Such a sensor which is combined with a polydimethylsiloxane (PDMS) wristband provides real-time information about the user's pulse and placed in PDMS glove comprehensive tactile feedback of a human hand touching. Some sensors used to detect human vital factors are created on electrospun nanofibers mats or on the non-textile flexible substrate such as Kapton [32]. Wearable sensors need also to be self-powered devices. It is possible to find the results of research on various energy harvesters or the triboelectric nanogenerators (TENGs). Parida et al [35] developed a 3D printable, highly conductive, extremely stretchable and healable TENG. The developed nanogenerator shows good conductivity (6250 S/cm) and record-high initial stretchability of 2500% and recovered 96.0% of its conductivity after healing. Some other examples of wearable energy harvesting devices constructed based on the thermoelectric phenomenon with desired dimensions are discussed in [34]. Another group worked on the sensors which monitored the perspiration lactate and illuminance and at the same time the energy harvester that extracted the electronic energy from the wearer's metabolic production and photoenergy from ambient illumination [36].

Sample refs to related work:

C. Irimia, M. S. Poboroniuc, S. Hartopanu, D. Sticea, G. Paicu and B. E. Ignat, "Post-stroke hand rehabilitation using a hybrid FES-robotic glove," 2016 International Conference and Exposition on Electrical and Power Engineering (EPE), Iasi, 2016, pp. 356-359.

Kattenstroth, J.C., Kalisch, T., Sczesny-Kaiser, M. Greulich W., Tegenthoff M., Dinse H. R.  Daily repetitive sensory stimulation of the paretic hand for the treatment of sensorimotor deficits in patients with subacute stroke: RESET, a randomized, sham-controlled trial. BMC Neurol 18, 2 (2018). https://doi.org/10.1186/s12883-017-1006-z

The mentioned references as the additional ones were added to the text. They could be found at the reference list as the 10,11 and 28-39  

All numbers of references have been corrected.

Attached please find the improved and corrected text of our article.

Round 2

Reviewer 2 Report

Paper and especially in troduction has been amended.

Still, the stroke affecting a large part of the society is surprising. I would prefer either numbers (percentages ...) or a more accurate sentence: the paper targets heavy consequences of stroke, and maybe all strokes do not have so heavy consequences (or are even not detected).

In te 1st sentences of the introduction :

"Stroke is a neurological disease that affects mainly elderly people. Currently, the number of people suffering from the first strokes has increased approximately by 68% and the number of the disabled people is also increasing largely in the world"

Again, more accurate data would be appreciable : 68% in how much time ? In Poland ? Europe ? USA ? world ?

Some light mistakes should be corrected prior to publication.

p9 l258 : legend in different page as the figure

The paper will then be OK for publication.

Author Response

Dear Reviewer,

We would like to thank you very much for your time you spent for correcting our paper and suggestions. They have helped us to improve the paper.

Below we have written answers for your comments. All of them are placed directly under the comments and they are highlighted in the text:

Paper and especially in Introduction has been amended.

Still, the stroke affecting a large part of the society is surprising. I would prefer either numbers (percentages ...) or a more accurate sentence: the paper targets heavy consequences of stroke, and maybe all strokes do not have so heavy consequences (or are even not detected).

In the abstract the additional information has been placed according to the reviewer’s suggestions:

“According to WHO data it is the second world’s biggest killer, accounting for near 6 million deaths in 2016 and it is about 30% of the total number of strokes per year. Other patients affected by such a disease should be rehabilitated as soon as possible. As a result of this phenomenon, paresis may occur.”

In the 1st sentences of the introduction :

"Stroke is a neurological disease that affects mainly elderly people. Currently, the number of people suffering from the first strokes has increased approximately by 68% and the number of the disabled people is also increasing largely in the world"

Again, more accurate data would be appreciable : 68% in how much time ? In Poland ? Europe ? USA ? world ?

The additional information has been placed in the mentioned sentence:

“Stroke is a neurological disease that affects mainly elderly people. Currently, the number of people suffering from the first strokes has increased approximately by 68% in Poland in the period time 1990 – 2016 and the number of the disabled people is also increasing largely in the world.”

Some light mistakes should be corrected prior to publication.

p9 l258 : legend in different page as the figure

the description under Figure 9 has been changed into: “The alternative electrodes placed on the patient’s hand.”

Attached please find the improved and corrected text of our article.